# Medicinal Plants Used for Anxiety, Depression, or Stress Treatment: An Update

**DOI:** 10.3390/molecules27186021

**Published:** 2022-09-15

**Authors:** Maša Kenda, Nina Kočevar Glavač, Milan Nagy, Marija Sollner Dolenc

**Affiliations:** 1Faculty of Pharmacy, University of Ljubljana, Aškerčeva Cesta 7, 1000 Ljubljana, Slovenia; 2Faculty of Pharmacy, Comenius University Bratislava, Odbojárov 10, 83232 Bratislava, Slovakia

**Keywords:** central nervous system, depression, anxiety, insomnia, medicinal plants

## Abstract

Depression, anxiety, stress, and other mental disorders, which are on the rise worldwide, are indications that pharmacological therapy can have serious adverse effects, which is why many patients prefer to use herbal products to treat these symptoms. Here, we reviewed plants and products derived from them that are commonly used for the above indications, focusing on clinical data and safety profiles. While lavender, hops, maypop, lemon balm, and valerian have consistently been shown in clinical trials to relieve mild forms of neurological disorders, particularly depression, anxiety, and stress, currently available data do not fully support the use of peppermint for anxiety disorders and depression. Recent studies support the use of saffron for depression; however, its toxicological profile raises safety concerns. St. John’s wort is effective in alleviating mild to moderate depression; however, careful use is necessary particularly due to possible interactions with other drugs. In conclusion, more studies are needed to validate the mechanism of action so that these plants can be used successfully and safely to alleviate or eliminate various mental disorders.

## 1. Introduction

According to the World Health Organization, the number of people suffering from depression and other mental disorders is increasing worldwide, especially in low-income countries, as life expectancy increases and more people reach the age at which these mental disorders normally occur [1]. Additionally, risk factors are more prevalent in these countries, i.e., poverty, unemployment, death from a close one, break-up, illness, mental stress, and alcohol and drug abuse. Globally, 300 million people (or 4.4%) have depression [1].

Mental health disorders are classified into depressive disorders and anxiety disorders [1]. These may present with different symptoms and last for months or years. They may be recurrent and severely affect the patient’s quality of life and ability to function. The cost of these conditions can be expressed in years of life with a disability. In 2015, anestimated 50 million years of disability were spent worldwide for depressive disorders and 24.6 million years for anxiety disorders [1]. In the same year, 788,000 people ended their lives [1].

The symptoms of depressive disorders are sadness, loss of interest or pleasure, feelings of guilt or low self-worth, disturbed sleep or appetite, feelings of tiredness, and poor concentration, which can lead to suicide [1]. They are divided into major depressive disorder or depressive episode and dysthymia. Major depressive disorder or depressive episode includes depressed mood, loss of interest and enjoyment, and decreased energy, and can be mild, moderate or severe. Dysthymia exhibits similar symptoms that are less intense but last longer [1].

The symptoms of anxiety disorders include feelings of anxiety and fear. Types of anxiety disorders are generalized anxiety disorder, panic disorder, phobias, social anxiety disorder, obsessive-compulsive disorder, and post-traumatic stress disorder [1]. Symptoms can be mild, moderate, or severe, and tend to be chronic.

Pharmacological therapy for depressive disorders uses tricyclic antidepressants, monoamine oxidase inhibitors, selective serotonin reuptake inhibitors, serotonin and norepinephrine reuptake inhibitors, norepinephrine and dopamine reuptake inhibitors, serotonin antagonist and reuptake inhibitors [2]. Pharmacological therapy for anxiety disorders includes selective serotonin reuptake inhibitors, selective serotonin and norepinephrine reuptake inhibitors, pregabalin, tricyclic antidepressants, buspirone, benzodiazepines, and monoamine oxidase inhibitors [3]. However, patients often do not adhere to these synthetic antidepressants or anxiolytic therapies due to adverse events or signatory delay in effectiveness.

Serious side effects of synthetic antidepressants and anxiolytics include headaches, sexual dysfunction, addiction, seizures, and suicide [4]. These were reduced in 45% of the studies, where herbal medicines were used for the same indications [4]. [5,6,7,8].

Plants and products derived from them that are commonly used in the Western world as dietary supplements or over-the-counter medicines for the above indications (the use of some of them is supported by the European Medicines Agency herbal monograph) were studied here, focusing on recent clinical trials, safety profiles and whether or not their use is justified.

## 2. Results

### 2.1. Hops (Humulus lupulus *L*.)

*Humulus lupulus* L., hops (Cannabaceae), is the most important and known species of *Humulus* [9]. It is native to central Europe and is industrially grown throughout the temperate regions of the North. This perennial and dioecious climbing plant can reach 10 m. A subject of scientific and industrial interest is female inflorescences (cones) consisting of leaf bracts and glandular trichomes in the lupulin glands, which contain essential oil (constituents: β-myrcene, β-caryophyllene, α-humulene, β-farnesene, α-selinene, β-selinene, humulene epoxides, β-bisabolol, 2-methyl-3-buten-2-ol, etc.), prenylated flavanones (isoxanthohumol = IX, 6-prenylnaringenin = 6PN, 8-prenylnaringenin = 8PN), prenylated acylphloroglucinols (humulone = HU, its derivatives, and lupulones), chalcones (xanthohumol = XH, desmethylxanthohumol), triterpenes, flavonols and tannins [10]. XH seems to be the main constituent in the glands, and IX, 6PN, and 8PN could be decomposition products that emerge during drying and storage [11]. 

Female inflorescence is important not only as a basic source for the brewing of beer but also for the preparation of herbal teas or herbal preparations (liquid and dry extracts and tinctures). The quality of the raw material is checked using methods described in a pharmacopoeial monograph [12]. These preparations are characterized as “traditional herbal medicinal products”, which can be used for the relief of mild symptoms of mental stress and to aid sleep as stated by the EMA in its monograph on hops [13].

This use can be attributed to the effects of the typical constituents of the hop cone (XH, IX, 8PN, and HU) as positive allosteric modulators of γ-aminobutyric acid (GABA) receptors, and GABA_A_ receptors [14,15]. Furthermore, XH is metabolized to IX conjugates as determined by the Legette group in a pharmacokinetic study in men (n = 24) and women (n = 24) using LC-MS/MS [16].

Although there are several clinical trials for mixed preparations of hops (especially with valerian) [17,18,19] only one study is available for single-component preparations [20] for the treatment of medical conditions directly related to the CNS. A group of 36 participants (females/males: 31/5; mean age: 24.7 years) was included in a randomized (1:1), placebo-controlled, double-blind, crossover design with two 4-week intervention periods (separated by a 2-week washout) with placebo or capsules (food supplement, two capsules of dry hop extract (0.2 g) once daily in the evening). Depression, anxiety, and stress symptoms were evaluated in all study participants using the Depression Anxiety Stress Scale-21 (DASS-21), which consists of 21 self-reporting items with seven items in each of the three subscales (depression, anxiety, and stress, respectively), documenting the relevant symptoms during the past week. A significant decrease in the anxiety, depression and stress was observed with both hops and placebo, which was greater with hops compared to placebo.

An interesting study is on the sedative effect of nonalcoholic beer in healthy female nurses. Overnight sleep and chronobiological parameters were assessed after moderate ingestion of 333 mL of beer sample for 14 days. The demonstrated improvement in sleep quality should be considered anecdotal, as it was assumed that nonalcoholic beer contained only 0.3% hops [21].

In conclusion, a limitation of only one “single-component” study is that the depression, anxiety, and stress symptoms of the participants were self-reported. Therefore, clinical trials of better quality are needed to determine the effect of hops on the CNS.

### 2.2. Kava-Kava (Piper methysticum *G. Forst*.)

Kava-kava, botanically known as *Piper methysticum* G. Forst., grows as a bush, to a height of 2 to 3 m. It is a dioecious plant. Its leaves are 13 to 28 cm long and 10 to 22 cm wide, with a deep cordate base and 9 to 13 main ribs and large stipules. The flowers are small, in spike-shaped inflorescences, 3 to 9 cm long. Kava-kava has a massive, branched rhizome and root system, weighing 2 to 10 kg. The plant is native to the Santa Cruz Islands and Vanuatu [22,23].

Part of the plant used is dried rhizomes of variable size. They can be 3–20 cm long and 1–5 cm wide [24]. The main components are 43% starch, 20% fibers, 3–20% kavalactones, 3.2% sugars, 3.6% proteins, 3.2% minerals (e.g., potassium, calcium, magnesium, sodium, aluminum, and iron), dihydrochalcones (e.g., flavokavins) and alkaloid pipermethystine [24]. Of these, the main active compounds are kavalactones. Several were found in a 95% ethanol extract: 11-hydroxy-12-methoxydihydrokavain; 7,8-dihydro-5-hydroxykavain; 11,12-dimethoxydihydrokavain; methysticin; dihydromethysticin; kavain; 7,8-dihydrokavain; 5,6-dehydromethysticin; 5,6-dehydrokavain; yangonin; 5,6,7,8-tetrahydroyangonin; 5,6-dihydroyangonin; 7,8-dihydroyangonin; 10-methoxyyangonin; 11-methoxyyangonin; 11-hydroxyyangonin; 5-hydrokavain; 11-methoxy-12-hydroxydehydrokavain. There are other compounds present in the 95% ethanol extract, which could also possess some activity: lavokavin A; flavokavin B; flavokavin C; dihydrokavain-5-ol; cuproic acid; cinnamalketone methylenedioxy-3,4-cinnamalketone; 4-oxononanoic acid; benzoic acid; phenyl acetic acid; dihydrocinnamic acid; cinnamic acid; pipermethystine; 1-(meta-methoxycinnamoyl)pyrrolidine and 1-cinnamoylpyrrolidine [24]. Different kava-kava extracts and some isolated compounds have been shown to interact with GABA_A_ receptors, inhibit monoamine uptake by inhibiting monoamine oxidase MAO-B, and modulate serotonin 5-HT_1A_ receptors [24]. In vivo experiments in rats and mice exhibited sedative, tranquillizing, and muscle relaxant properties of different kava-kava extracts and some isolated compounds [24]. In addition, some studies showed anticonvulsive, spasmolytic, neuroprotective, and analgesic activities [24].

Clinical trials showed different results on the effectiveness of kava-kava preparations in anxiety disorders. A randomized double-blind controlled trial that included 135 participants in the kava-kava group and 135 participants in the placebo group showed improvements in anxiety symptoms and sleep, but there were no differences between the test groups, so kava-kava did not improve those symptoms more than the placebo [25]. Another study analyzed a pool sample of three randomized double-blind controlled trials and observed no improvement in the kava-kava treatment group [26]. Furthermore, no hepatotoxicity has been observed in the kava-kava treatment group. In contrast, a randomized double-blind controlled study by Sarris et al. showed a significant reduction in anxiety in the kava-kava treatment group, and this effect was more pronounced in individuals with moderate to severe generalized anxiety disorder [27]. Polymorphisms in GABA transporters rs2601126 and rs2697153 were associated with this outcome. More headaches were present in the kava-kava treatment group, while no differences in liver function tests were seen between treatment groups. Overall, clinical trials on the effectiveness of kava-kava in generalized anxiety or anxiety in (peri)menopause, which were reviewed by the European Medicines Agency in 2016, were found to have major shortcomings [24]. These were short duration of trials and the follow-up phase, a not uniform anxiety population, not enough information on the percentage of responders, different extraction methods of supplements used, different reference compounds and different dosages used [24].

Kava-kava extracts have been discontinued in some countries, as they were hepatotoxic [28]. Spontaneously reported liver adverse reactions, including those of such severity that liver transplants were required, were one of the reasons that risk-benefit balance was considered unfavorable and led the European Medicines Agency to the decision to not establish an herbal monograph for kava-kava [29]. In addition, the carcinogenic potential in animals raised toxicological concerns [29].

The methodological weaknesses of clinical trials that demonstrate the effectiveness of kava-kava in anxiety disorders and toxicological concerns are the reason why this plant cannot be considered effective or safe for this indication.

### 2.3. Lavender (Lavandula angustifolia *Mill*.)

*Lavandula angustifolia* Mill. is an aromatic medicinal plant native to areas that extend from Italy, France, and Spain. It typically grows from 0.5 to 1 m. The leaves are haired and lanceolate with a decussate leaf arrangement. The flowers have a violet to violet-blue color and a form characteristic of the Lamiaceae family, i.e., with petals fused into an upper lip and a lower lip, symmetric. Individual flowers appear at the top of a stem, in clusters of 6 to 10 [30,31].

Lavender flowers are the herbal material used for the following indications: relief of mild symptoms of mental stress and exhaustion, and to aid sleep according to the European Medicines Agency monograph on lavender [32]. These approved therapeutic indications are based on traditional use. Lavender flowers can be used orally as tea or for the preparation of a tincture and essential oil obtained by steam distillation [32,33]. Essential oil can also be used as a bath additive [33]. 

Lavender flowers contain 1–3% essential oil, coumarin derivatives (umbelliferon, herniarin), flavonoids, traces of sterols (cholesterol, campesterol, stigmasterol, β-sitosterol), traces of triterpenes (mictomeric acid, ursolic acid), up to 13% tannins, phenolcarboxylic acids (e.g., rosmarinic acid, ferulic acid, isoferulic acid, α-coumaric acid, *p*-coumaric acid, gentisic acid, *p*-hydroxy-benzoic acid, caffeic acid, melilotic acid, sinapic acid, syringic acid and vanillic acid) [34]. Lavender oil contains 60–65% monoterpene alcohols (e.g., linalool, linalyl acetate, *cis*-ocimen, terpinene-4-ol, limonene, cineole, camphor, lavandulyl acetate, lavandulol and α-terpineol, β-caryophyllene, geraniol, α-pinene), non-terpenoid aliphatic compounds (e.g., 3-octanone, 1-octen-3-ol, 1-octen-3-ylacetate, 3-octanol) [34]. In vitro, lavender oil has antimicrobial, spasmolytic, as well as estrogenic activities [34]. In vivo in rats and mice, anticonvulsive, sedative, anti-inflammatory, and analgesic effects have been observed [34]. As these studies generally used high doses, these effects may not be reflected in humans at relevant doses.

Few controlled clinical trials have shown that oral lavender oil helps with symptoms of generalized anxiety disorder and mixed depression/anxiety symptoms. A randomized controlled double-blind clinical trial included 539 patients with generalized anxiety disorder [35]. Participants received lavender oil preparation, paroxetine, or placebo for 10 weeks. Lavender oil preparation decreased the Hamilton anxiety scale score by more than 50% in 60% of the treated patients, while the incidence of adverse events was comparable to the placebo group. A similar study design addressed the effectiveness of lavender oil preparation in anxiety-related restlessness and disturbed sleep [36]. It was found that 48.8% of the patients responded to treatment compared to 33.3% in the control group. In a randomized placebo-controlled study, the lavender oil treatment group had significantly better outcomes for mixed anxiety and depressive disorder than the placebo group [37]. When lavender oil preparation was compared to lorazepam treatment in patients with a generalized anxiety disorder during a randomized controlled clinical 6-week trial, lavender oil preparation performed similarly to lorazepam treatment, without having a sedative effect or addiction potential [38]. As some studies comparing the effectiveness of lavender oil and synthetic antidepressants/anxiolytics used insufficient doses of the latter, more studies are needed to fully elucidate the effectiveness of oral lavender oil compared to therapy with synthetic agents [28]. However, based on an extensive review of all relevant clinical trials by the European Medicines Agency in 2010, it was concluded that lavender oil seems to help with anxiety and stress-induced restlessness but the criteria for a well-established use are not met [34].

Lavender is contraindicated if hypersensitivity to the herbal substance is present, for example, pollen allergy [32]. Lavender oil allergy-induced dermatitis has been reported [34]. The topical use of products containing lavender oil has been associated with gynecomastia in three prepubertal boys [34]. Lavender oil baths are contraindicated if there are open wounds, skin injuries/diseases, high fever, severe infections, or severe circulatory and cardiac problems present [33]. No interactions with drugs or other interactions have been reported. Due to its sedative effect, lavender might compromise the ability of some patients to drive and operate machinery.

Although there is a lack of firm evidence for the use of lavender flowers and lavender oil for anxiety and related sleep disorders, positive effects have been reported. Therefore, the European Medicines Agency included such indications for the use of lavender oil based on traditional use [34]. The latter combined with a good toxicological profile makes lavender oil one of the options that are worth trying if experiencing mild anxiety and consequent sleep problems.

### 2.4. Lemon balm (Melissa officinalis *L*.)

Lemon balm or common balm, with the scientific name *Melissa officinalis* L., family Lamiaceae, is a perennial that grows up to 90 cm in height. The stem is quadrangular, sparsely haired to glabrous. The leaves are petiolate, ovate to deltoid-ovate, 2 to 6 cm long and 1.5 to 5 cm wide. Bilabiate flowers are white to pink in color. The plant originates in the Mediterranean and west Asian areas [22,39,40].

The part of the plant used for medicinal purposes are the leaves. They can be used as an herbal substance in herbal teas, or made into other dosage forms (comminuted or powdered herbal substance, ethanol liquid extract, tincture, or dry extracts) [41]. The European Medicines Agency approved lemon balm for the relief of mild symptoms of mental stress and to aid sleep, as well as for certain gastrointestinal problems [41]. Both indications were granted based on traditional use [41]. The dried lemon balm leaves contain 0.06-0.8% essential oil (which contains monoterpene aldehydes, mainly citral, neral and citronellal), sesquiterpene derivatives (β-caryophyllene and germacrene D), monoterpene glycosides, flavonoids (luteolin, quercetin, apigenin, and kaempferol glycosides), tannins, triterpenes (such as ursolic acid and oleanolic acid) [41]. In a study of rat brain homogenate, aqueous lemon balm extract exhibited inhibition of GABA transaminase activity, leading to increased levels of GABA, which could contribute to anxiolytic activity [42]. Ethanolic lemon balm extract was shown to bind to cholinergic receptors (muscarinic and nicotinic receptors) [43]. Compounds of the essential oil were shown to bind to GABA_A_ receptors [44]. In vivo studies showed increased GABA levels, proliferative and neuroblast differentiation effects, and sedative, narcotic, and anxiolytic effects when administered orally to mice [41].

The European Medicines Agency revised the assessment report on lemon balm in 2012. Here, we summarize clinical trials on the effectiveness of lemon balm in anxiety disorders, which were included in the assessment report, as well as some additional studies published since. A study by Kennedy et al. included 20 healthy volunteers who received 300, 600, or 900 mg of lemon balm extract daily [45]. Self-reported calmness increased at 1 to 2.5 h after the 300 mg dose, while secondary and working memories decreased at 2.5 to 4 h after the 600 and 900 mg doses. In a similar study, the 600 mg dose helped with the reaction to experimentally induced stress, increased calmness, and decreased alertness, while the 300 mg dose accelerated mathematical processing [46]. In a randomized placebo-controlled clinical trial in Alzheimer patients, 16-week administration of lemon balm ethanolic extract was found to have a better effect on cognitive functions than placebo [47]. Another randomized, placebo-controlled trial in Alzheimer’s patients showed superior effects of lemon balm aromatherapy on agitation compared to donepezil or placebo [48]. Twenty volunteers who took 30% lemon balm extract, containing at least 7% rosmarinic acid and 15% hydroxycinnamic acid derivatives, for 15 days reported reduced anxiety symptoms and insomnia [49]. However, no control group was included in this study. Lemon balm aromatherapy in 72 patients who came to the emergency department with the acute coronary syndrome had beneficial effects on the mean stress score, heart rate, and mean arterial pressure [50]. Aqueous lemon balm dry extract, taken for 14 days in a daily dose of 500 mg, helped with heart palpitations and anxiety in a randomized placebo-controlled trial in 71 eligible volunteers [51]. In a randomized placebo-controlled clinical trial, 80 patients with stable chronic angina were taking 3 g of lemon balm supplement for 2 months [52]. Significant reductions in anxiety, stress, depression, and improvement in sleep disturbances were reported in the treatment group compared to the placebo group. A recent (2021) meta-analysis concluded that evidence from randomized clinical trials suggests the effectiveness of lemon balm supplements in improving anxiety and depression symptoms, especially those of acute nature [53]. However, the studies considered were heterogenic and had shortcomings; therefore, the results of this meta-analysis should be interpreted with caution.

Hypersensitivity to the herbal substance and impaired ability to drive and operate machines may be present in some individuals [41]. No interactions with drugs have been reported.

Clinical trials do not preclude traditional indications for use, but these studies are not sufficient to justify well-established indications for use. Because there are no significant concerns with the use of lemon balm and some data support its antianxiety effects, lemon balm could be used to relieve mild anxiety symptoms and support sleep.

### 2.5. Maypop (Passiflora incarnata *L*.)

*Passiflora incarnata* L., family Passifloraceae, is commonly known as maypop or the true passionflower. It is a perennial vine native to the southeast United States, Argentina, and Brazil. The stem can reach up to 10 m. The leaves are alternate, petiolate, serrate, and finely pubescent, with denser hairs on the underside. The flowers have a diameter of up to 9 cm, petals and sepals are white, the outer corona is purple or pink, and the inner corona is white and shorter. Fruits are yellow, ellipsoid, 5 cm in diameter and edible [22,54].

The phytochemicals found in aqueous extracts of maypop are *C*-glycosylated flavonoids [55], such as vitexin, isovitexin, schaftoside, isoschaftoside, orientin, isoorientin, and swertisin. Furthermore, the free flavonoids apigenin, luteolin, quercetin, kaempferol, and chrysin are also found [56]. The alkaloids present in *Passiflora* species are of the indole type (β-carbolines) and have been shown to be effective as sedatives and to lower blood pressure. Maypop is the most studied among *Passiflora* species that contain alkaloids [57]. Lutomski et al. reported in the 1960s the occurrence of harmine, harmol, harmaline, harmalol, and harman in the raw *Passiflora* material [58]. The presence of these alkaloids has been confirmed, but only in traces, so they cannot be detected in most commercially available materials [55].

The chemical constituents responsible for the anxiety-relieving effects of maypop are not yet fully understood, but most of the published works suggest that phenolic substances, primarily from the flavonoid class, are associated with this property. The mechanism of action is probably related to the modulation of the GABA system, because *Passiflora* flavonoids are partial agonists of GABA_A_ receptors and inhibit the uptake of [^3^H]- GABA in rat cortical synaptosomes [55]. Wasowski and Marder describe flavonoids as GABA_A_ receptor ligands, including apigenin and chrysin, that bind to the benzodiazepine binding site of the GABA_A_ receptor and exhibit anxiolytic activity without showing sedative and muscle relaxant effects [59]. The harman alkaloids found in maypop have different structural characteristics that interact with benzodiazepine receptors. In addition, research involving β-carboline compounds has led to conceptual and experimental considerations covering the role of ligands in the GABA_A_ receptor modulation process in a spectrum ranging from full agonists to full inverse agonists [60]. Therefore, many of the pharmacological effects of maypop are mediated by modulation of the GABA system, as β-carboline compounds have shown an affinity for the GABA_A_ and GABA_B_ receptors, and effects on GABA uptake. Aman et al. conducted a study in mice that showed that maypop could be useful in the treatment of neuropathic pain [61,62]. The authors suggest that these properties are due to the modulation of opioid and GABA-ergic mechanisms, but also point to the possible involvement of oleamides as cannabimimetics. Janda et al. included nine clinical trials in their systematic review of maypop on the nervous system [63]. The duration of the studies included in the analysis varied widely, from one day to 30 days. Study participants were not less than 18 years of age. In each of the studies, the effects of passionflower were measured using various tests and scales. In most of the studies, a reduction in anxiety was observed after taking maypop preparations, although the effect was less marked in people with mild anxiety symptoms. No adverse effects such as memory loss or changes in psychometric functions were observed.

Maypop is used mainly in combination products with hops and valerian; however, it can also be found without other plant extracts even as over-the-counter drugs. Safety and toxicity data are not available. Although maypop is not known to contain toxic compounds and no side effects have been reported, long-term use (> 4 weeks), use during pregnancy and lactation, or in children/adolescents under 12 years of age cannot be recommended, as stated by the European Medicines Agency in its maypop monograph [64]. As a sedative, maypop can affect the ability to drive and operate machinery.

### 2.6. Peppermint (Mentha × piperita *L*.)

*Mentha* × *piperita* L., peppermint, is a sterile hybrid between *Mentha aquatica* L. and *Mentha spicata* L. The plant is one of the most widespread plants in the Lamiaceae family, native to Europe, Turkey, and several parts of west Asia. It is a perennial that grows up to 90 cm. The stem is quadrangular, and the leaves are aromatic, petiolate, oblong-ovate, and serrate, with a decussate leaf arrangement. The flowers are bilabiate and of violet color [22,65,66].

The part of the drug used is the leaf. Dried leaves are used as a herbal substance, they can be comminuted or used in the form of 45% and 70% tinctures, or peppermint essential oil (obtained by steam distillation of the plant at the flowering stage) [67]. A tea, solid, or liquid dosage forms may be prepared thereof [68]. The entire herbal substance contains at least 12 mL/kg of essential oil, and the cut herbal substance contains at least 9 mL/kg of essential oil [67]. The leaves contain fatty acids (palmitic, linoleic, linolenic acid), flavonoids (e.g., luteolin, its 7-glycoside, rutin, hesperidin, eriocitrin), phenolic acids, and triterpenes [67]. Essential oil is mainly composed of up to 55% menthol and up to 32% menthone [67]. Other constituents are limonene, cineole, menthofuran, isomenthone, methyl acetate, pulegone, carvone, and isopulegol [67]. There is non-clinical evidence that peppermint has antispasmodic, choleretic, antinociceptive, anticarminative, bronchomucotropic, antioxidant, antimicrobial, antiplasmid, antiviral, antiallergic, central nervous system, anticonvulsant, diuretic, chemoprotective, renoprotective, and hepatoprotective activities [67]. Compounds of peppermint oil and menthol are believed to bind to the serotonin 5-HT_3_ receptor and were able to reduce serotonin-induced ileum contractions [69]. Sedative effects on induced sleep, behavior, motility, and coordination were observed in mice treated with aqueous peppermint extract [70].

Data on the use of peppermint in anxiety disorders and depression are limited. The European Medicines Agency did not include these indications in the assessment report based on which the following indications were approved in the herbal monograph: mild gastrointestinal cramps, flatulence, abdominal pain, dyspepsia, mild headache, cough and cold, localized muscle pain, and localized itching conditions with intact skin [68,71]. However, few studies have addressed the effects of peppermint on anxiety and depression. A study by Adam et al. showed that peppermint odor reduced fatigue and improved mood and sleep [67]. A recent (2021) study by Abdelhalim et al. assessed the effect of peppermint infusion and fresh parts on the mental health of university students [72]. Students in the treatment group reported better memory function and quality of sleep and reduced anxiety. The outcomes were significantly better than in the control group, however, this study was not blind. Another recent study (2020) reported the beneficial effects of 7-day peppermint oil aromatherapy on sleep quality scores in cancer patients [73]. In contrast, the smell of peppermint can also be perceived as stimulating. A study by Goel and Lao showed that the perception of peppermint odor as more intense is correlated with greater total sleep and slow-wave sleep, while interindividual differences also played an important role [74]. Peppermint oil has also been shown to improve memory function and increase alertness, as well [75]. A 2019 randomized controlled trial evaluated the effect of peppermint oil aromatherapy on pain and anxiety due to intravenous catheterization [76]. Reduced pain and anxiety scores were observed in the treatment group compared to the control group. However, the study was not blind, as the peppermint odor was recognizable in the treatment group and no odor was present in the control preparation. Furthermore, the groups of patients were of heterogeneous age, which could affect the perception of pain.

There are some safety concerns with peppermint use. It is contraindicated in patients with hypersensitivity to peppermint or menthol [71]. It should not be used before the age of two, as menthol can cause reflex apnea and laryngospasm [71]. Peppermint essential oil is also contraindicated in patients with biliary disorders or liver disease, cholangitis, and achlorhydria, and in children who had suffered from seizures [71]. It is important to note that peppermint essential oil should be ingested in gastro-resistant dosage forms to avoid irritation of the mouth and esophagus [71]. Concomitant use of antacids should be avoided to prevent premature release from gastro-resistant dosage forms. In patients who experience heartburn or who have previously had heartburn, symptoms may worsen. In this case, patients should stop taking peppermint oil. Eye contact with essential oil should be avoided, as it can cause eye irritation [71]. The oil should not be applied to irritated skin. Adverse reactions to peppermint essential oil include bronchoconstriction and laryngoconstriction when inhaled by hypersensitive individuals, allergic reactions (such as anaphylactic shock, skin rash, contact dermatitis), headache, bradycardia, tremor, heartburn, blurred vision, dry mouth, nausea, vomiting, dysuria, inflammation of the glans penis [71]. No additional ingestion of peppermint is recommended on top of the treatment regimen. Overdose with oral ingestion of peppermint essential oil is possible and includes severe gastro-intestinal symptoms, diarrhea, rectal ulceration, epileptic convulsions, loss of consciousness, apnea, nausea and disturbances in cardiac rhythms, ataxia, and other CNS problems [71]. The stomach should be emptied if this occurs. Overdose by inhalation of peppermint essential oil is also possible and includes symptoms such as dizziness, confusion, muscle weakness, nausea, and double vision [71].

In general, insufficient evidence was found to fully support the use of peppermint in anxiety disorders and depression. Some studies show beneficial effects but have significant shortcomings. Adverse effects are possible and frequent with the use of peppermint, especially when essential oil from peppermint is ingested.

### 2.7. Saffron (Crocus sativus *L*.)

Saffron, botanically termed *Crocus sativus* L., family Iridaceae, originates in Greece, from where it was introduced to former Czechoslovakia, Italy, Spain and Morocco, and Turkey, Iran, Pakistan, and the West Himalayas. It can grow up to 30 cm in height. It has a corm, 3 cm in diameter. The leaves are erect or splayed, up to 20 cm long and 2 to 3 mm wide. The flowers are 1 or 2, with a violet calyx, yellow anther, and an orange stigma [22,77,78].

Saffron has been used in traditional medicine and is one of the most expensive spices [79]. It is made up of 63% sugars, 12% protein, 10% water, 5% fat, 5% crude fiber, and 5% minerals [80]. The part used in phytotherapy is dried stigma. It contains flavonoids, vitamins, and carotenoids [81]. Active ingredients with antioxidant activity are, for example, the apocarotenoids crocin, picrocrocin, and safranal, which can scavenge radicals in addition to modulation of enzymes that combat oxidative stress, pro-apoptotic effect, decreased synthesis of DNA, RNA, and proteins, and decreased telomerase activity [81]. These same ingredients give saffron its distinct color, taste, and odor [80]. In terms of antidepressant activity, crocin, crocetin, and *N*-acetylcysteine have been recognized to help alleviate symptoms of depression [82]. Mechanisms responsible for antidepressant activity may involve the opioid system and the GABAergic system (via GABA_A_ receptors) [83,84]. Animal experiments showed antidepressant activity with increased levels of brain-derived neurotrophic and nerve growth factors [80].

A 2013 meta-analysis showed that saffron can improve major depressive disorder and a 2014 systemic review showed that saffron had antidepressant activity similar to synthetic antidepressants [85,86]. A recent meta-analysis, published in 2019, has provided an update on the use of saffron for mild to moderate depression compared to placebo and synthetic antidepressant treatments [82]. Eleven randomized controlled clinical trials have been considered, of which nine were included in the statistical analysis, which showed that oral administration of saffron preparations in pharmacological doses ameliorates depression symptoms compared to the placebo group and is comparable to treatment with synthetic antidepressants.

A typical dose in clinical trials was 30–200 mg of saffron extract per day. In terms of clinically observed safety, no significant adverse events were reported in these studies [87,88,89,90]. The acute lethal dose of saffron after oral administration is 4.1 g/kg in mice [91]. The teratogenic effects of saffron or its components have been reported in animal studies [91]. Increased miscarriage rates were reported among pregnant saffron field workers [92]. Therefore, it is not recommended to take saffron supplements during pregnancy.

Although novel data from recent studies support the use of saffron in depression, its toxicological profile raises safety concerns. However, based on the doses used in the clinical trials conducted so far, saffron could help alleviate depression and does not cause significantly more adverse effects in nonpregnant individuals than without treatment.

### 2.8. St. John’s wort (Hypericum perforatum *L*.)

*Hypericum perforatum* L., commonly known as St. John’s wort, belongs to the family of Hypericaceae. It originates from the temperate areas of Eurasia. The perennial reaches a height of up to 100 cm. It has a reddish, branched stem, and lanceolate to elliptic, translucent leaves, with a perforated appearance; hence the plant species name, *perforatum*. The leaf margin is covered with translucent “spots”, which are glands that produce hyperforin. The flowers are golden yellow and grow in terminal cymes. At the petal margins, there are naphthodianthrone-producing glands that give the characteristic appearance of black “spots” [22,93,94].

The part of the plant used for medicinal purposes are whole or cut flowering tops [95]. St. John’s wort contains phloroglucinol derivatives (hyperforin, adhyperforin, furanohyperforin), naphthodianthrones (pseudohypericin, hypericin, protohypericin, protopseudohypericin, cyclopseudohypericin, skyrin derivatives), flavonoids (glycosides of quercetin, e.g., hyperoside, rutin, isoquercitrin, quercitrin, biflavones, I3,II8-biapigenin, amentoflavone), procyanidins (procyanidin B2, tannins with catechin skeleton), xanthones, essential oil (contains compounds 2-methyl octane, α-pinene, caryophyllene, geranyl acetate, and nonane), cholinergic acid, caffeoylquinic, *p*-coumaroylquinic acids and free amino acids [95]. The dried herbal substance contains at least 0.08% total hypericines (expressed as hypericin) and is used for the preparation of dry extracts with 80% v/v methanol, 80% *v*/*v* ethanol, or 50–68% *v*/*v* ethanol as extraction solvents, as defined in the monograph of the European Medicines Agency [96]. Such dry extracts inhibit the reuptake of noradrenaline, serotonin, and dopamine, as well as downregulate the β-adrenergic receptors, which resembles the mode of action of synthetic antidepressants [96]. Active compounds are naphthodianthrones (hypericin and pseudohypericin), phloroglucin derivatives (hyperforin), and flavonoids [96].

In 2009, the European Medicines Agency published an assessment report on St. John’s wort, including antidepressant activity, anxiolytic, neuroprotective, memory and nootropic effects [95]. The approved therapeutic indications are mild to moderate depressive episodes and short-term treatment of symptoms in mild depressive disorders [96]. Daily doses range from 500 to 1800 mg and result in a therapeutic effect within 4–6 weeks [92]. Since then, few clinical trials have been conducted on the effects of St. John’s wort on the central nervous system. A large meta-analysis, published in 2017, considered 27 clinical trials to determine the effect of St. John’s wort on depression versus selective serotonin reuptake inhibitors [97]. The results were very encouraging, as the response and remission rates were comparable to selective serotonin reuptake inhibitor therapy, while the dropout rate was significantly better in St. John’s wort treatment. However, none of the clinical trials considered lasted more than 12 weeks, making it impossible to determine long-term effectiveness and benefits for patients with severe depression or suicidal patients. In addition, specific safety concerns arise with the use of this plant.

St. John’s wort is known to interact with many drugs. Since it induces the activity of CYP3A4, CYP2D9, CYP2C19, and the P-glycoprotein, it is incompatible with some drugs and caution should be exercised, as the concomitant use of St. John’s wort could render the drugs toxic or ineffective [96]. This could lead to a worsening of the patient’s disease status or result in an unplanned pregnancy. St. John’s wort is contraindicated with cyclosporin, tacrolimus, amprenavir, indinavir (as well as other protease inhibitors), irinotecan, and warfarin. Caution should be exercised if used in conjunction with amitriptyline, fexofenadine, benzodiazepines, methadone, simvastatin, digoxin, finasteride, and oral contraceptives. The induced activity of the aforementioned enzymes is restored to normal levels in one week after stopping the St. John’s wort therapy or supplementation. This may be advised in some cases to avoid interactions, for example, before undergoing anesthesia. Furthermore, St. John’s wort might have serotonergic effects in combination with some types of antidepressants, namely selective serotonin reuptake inhibitors [96]. One component of St. John’s wort, hypericin, is phototoxic and can cause sunburn-like symptoms; therefore, sun exposure or cosmetic laser treatments should be avoided if taking St. John’s wort preparations [98,99,100]. Due to insufficient clinical data, it is not recommended to take St. John’s wort during pregnancy or lactation [96]. Allergic skin reactions can also occur. Other side effects of St. John’s wort-containing preparations are gastrointestinal disorders, fatigue, and restlessness [96]. A case of overdose reports seizures and confusion after ingesting 4.5 g of dry extract daily for 2 weeks followed by a 15 g single-dose ingestion [96].

In conclusion, St. John’s wort is effective in ameliorating depression and its use is beneficial in patients with mild to moderate depression [97]. Due to many interactions with other drugs and possible adverse effects, patients must be properly educated before using St. John’s wort preparations.

### 2.9. Valerian (Valeriana officinalis *L*.)

The *Valeriana* genus, family Caprifoliaceae, contains 289 species [101]. The most important is *Valeriana officinalis* L. (Valerian) including the subspecies *Valeriana officinalis* subsp. c*ollina* (Wallr.) It is native to Europe and western Asia. It can be found in both moist and dry locations. From the point of view of morphology, valerian is a very diverse plant. In the second year of growth, the plant forms a round, furrowed, and hollow flower stalk, 80 to120 cm high and branched at the top. The pale green (above) lanceolate, pinnate leaves grow from either a pinnate form or 9 to 21 finely toothed leaflets. Leaves are attached in pairs on either side of the stem. The stems end in umbels that bear many branches and tiny white and pale pink flowers. Valerian has a rhizome and a root with many secondary roots and stolons. The part of the plant of interest is a rhizome with fascicled roots, 1.5–2.5 mm in diameter [102].

As the rhizome and roots of valerian are an important pharmaceutical source, the European Pharmacopoeia requires quality control for dried, whole, or fragmented underground parts [12].

Dozens of clinical trials with valerian root preparations containing its aqueous extract, ethanolic extract, or comminuted plant material began to be published in the early 1980s. Their results confirmed that aqueous ethanol extracts have a clinical effect on sleep disorders, especially in elderly patients. Typical constituents are iridoids, flavonoids, and essential oil containing monoterpenes and sesquiterpenes. Clinical observations indicated that for the treatment to have the expected effect, it must last several weeks. All these occupations were discussed in detail in an assessment report of the European Medicinal Agency [103], which was the basis for the development of a community monograph on valerian root, which lists the category “well-established use” the indication “for the relief of mild nervous tension and sleep disorder” and for the category “traditional use” the indication “for the relief of mild symptoms of mental stress and to aid sleep” [104]. Thus, both indications are clearly related to the in vitro effects on the GABAergic system described above.

Clinical trials that were published after the above assessment report or were not included in it can be divided into two groups: (1) those containing comminuted root and (2) those based on the aqueous ethanol extract. In a non-controlled case study [105], 16 female and 4 male Hispanic volunteers, ages 43 to 72 years, completed the two-week trial. Eleven persons had a diagnosis of major depression according to the “Diagnostic and Statistical Manual of Mental Disorders-4”, and two had a primary sleep disorder. Patients were asked to continue all concomitant medications, including all sedatives, at their baseline dose during the 2-week trial. They were encouraged to abstain from alcohol and street drugs, as well as from all caffeinated beverages after 5:00 p.m. and instructed to take one capsule of valerian root (470 mg) each night, 30–60 min before retiring. All volunteers increased their dose to three capsules during the second week of the trial. The results of the global ratings for week 2 showed that Valerian‘s subjective hypnotic effect was superior to that of week 1. Most of the patients (80%) again reported that valerian was at least “moderately” helpful. However, 30% thought it was “extremely” helpful. No side effects were attributed to valerian by any of the patients.

A prospective, triple-blinded, randomized, placebo-controlled, parallel design study [106] was used to compare the effectiveness of valerian capsule (400 mg dry root, 0.58 mg of valerenic acid content) with placebo on sleep quality and severity of symptoms in patients (average age 50 years) with restless legs syndrome (RLS). Most subjects had severe (38.9%) or very severe (19.4%) RLS symptoms at enrollment, as inclusion criteria required symptoms occurring three times a week. Thirty-seven randomly assigned participants completed the study (valerian group: 12 women and 5 men, placebo group: 15 women and 5 men). They received two valerian capsules or a placebo for eight weeks. Sleep quality was evaluated using the Pittsburgh Sleep Quality Index (PSQI) and sleepiness using the Epworth Sleepiness Scale (ESS). All patients experienced an improvement in sleep quality and RLS severity over the course of the study. PSQI scores decreased for all components with a significant (*P* < 0.05) decrease for subjective sleep quality, sleep latency, sleep duration, habitual sleep efficiency, and sleep disturbance. ESS also showed improvement in all subjects.

Pinheiro et al. [107] evaluated the effectiveness of valerian root (single oral dose of 100 mg 1 h before a surgical procedure) for the control of anxiety during third molar surgery in 20 volunteers (12 women and 8 men, age range between 17 and 31 years). The degree of anxiety of the patient was assessed using questionnaires and physical parameters (heart rate and systolic and diastolic blood pressure). Taking into account the limitations of the present study (sample size calculation was not performed, patients both anxious and non-anxious (determined by the DAS scale) at the first appointment were included in the research), valerian root capsule administered had a greater anti-anxiety effect than placebo.

A randomized, double-blinded, placebo-controlled trial was conducted with 61 patients of both sexes, aged 30 to 70 years, who are candidates for coronary artery bypass graft surgery using cardiopulmonary bypass [108]. Cognitive brain function was evaluated prior to surgery and at 10 days and 2 months of follow-up by the Mini-Mental State Examination test. Two groups of participants received valerian capsules (530 mg of valerian root) or a placebo, every 12 h. The intake of valerian and placebo began 1 day before surgery and continued 60 days after surgery. This resulted in reduced odds of cognitive dysfunction in the valerian group compared to the placebo group, which could be related to improvement in sleep quality.

In a pilot randomized, double-blinded, placebo-controlled, clinical trial [109], 51 HIV-positive patients (17 women and 24 men) who received efavirenz were recruited into the valerian (n = 25, mean age 36 years) or placebo (n = 26, mean age 34 years) group. Their neuropsychiatric parameters (sleep, anxiety, depression, suicidal thought, and psychosis) were evaluated at week 0 and week 4 using validated questionnaires (Hamilton Depression Rating Scale, Hamilton Anxiety Rating Scale, Positive and Negative Syndrome Scale, Positive and Negative Suicide Ideation, and Pittsburg Sleep Quality Inventory). The patients in the valerian group received one capsule (530 mg of valerian root powder) every night for 1 h before going to sleep. Valerian preparation significantly improved sleep and anxiety, suicidal thoughts improved insignificantly, and no change in psychosis symptoms was detected. In analyses between groups, sleep and anxiety improved significantly in the valerian group compared to the placebo group.

Pakseresht et al. recruited 31 adult patients who met the criteria for obsessive-compulsive disorder for a double-blind and randomized eight-week trial. Persons in the valerian group (eight women and seven men, mean age 31 years) or placebo one (seven women and nine men, mean age 29 years), respectively, received an oral capsule three times a day (250 mg of aqueous valerian root extract versus a placebo capsule) [12,110]. Effectiveness was evaluated using the Yale–Brown Obsessive Compulsive Scale. The valerian extract had some anti-obsessive and compulsive effects; however, the difference between the extract and placebo at the end of treatment was significant.

The objective of the randomized, placebo-controlled, double-blind, cross-over study of the Thomas Kelan group [111] was to determine the effects of a single dose on subjective sedation effects, standardized field sobriety test, and driving simulator performance parameters in 40 healthy adult participants (mean age 28.3 years) (24 women). The study consisted of two separate sessions of 2 to 3 h. In each session, participants received a dose of valerian (1600 mg of root extract (extractant not specified) containing 0.8% valerenic acid) or placebo and waited for 1 h for absorption. There were no differences between exposure conditions on the median scores of the Karolinska Sleepiness Scale, with subjects feeling “alert” for placebo exposure and “rather alert” for valerian exposure. There were also no differences in the mean scores of the Stanford Sleepiness Scale, with subjects feeling that they were “relaxed, awake but not fully alert, responsive” for valerian exposure and “functioning at a high level, but not at peak and still able to concentrate” for placebo exposure. There were no differences between valerian and placebo exposures on mean reaction time in the “simple visual reaction time” test or in “driving simulator performance”, as well as in “standardized field sobriety testing”. Therefore, a one-time dose of valerian 1600 mg is not expected to cause a failure of the field sobriety test or impaired driving performance.

Roh et al. [112] conducted a four-week, double-blinded, randomized, placebo-controlled clinical trial with 64 volunteers suffering from psychological stress. They received capsules (100 mg of a valerian root extract containing 0.8% valerenic acid) or placebo three times a day. The effects on anxiety and stress-related psychological constructs (EEG coherence in the alpha and theta frequency bands) were evaluated. The valerian root sample and the placebo groups both showed significant post-intervention improvements on all clinical scales. Compared to the placebo group, the “valerian” group exhibited significantly higher increases in alpha coherence of the frontal brain region, which was significantly correlated with anxiolysis [18].

Fifteen healthy right-handed college students (nine men and six females; mean age of 30 years) participated in a randomized, double-blind, cross-over study with an unspecified valerian extract containing 0.8 mg of valerenic acid [113]. Participants were required to take three capsules (900 mg extract in total) under two drug conditions, separated by 3 weeks. Then, several parameters of corticospinal excitability were investigated. The authors provided evidence that acute administration of valerian extracts affected motor cortex excitability with a decrease in intracortical facilitation, which was reversible. It is possible that valerian extract and valerenic acid allosterically modulate GABA_A_ receptors and, in this way, are thought to induce anxiolytic activity.

A prospective, randomized, split-mouth, crossover, double-blind study included 20 patients (11 women and 9 men, mean age of 23 years) with an indication for bilateral extraction of the mandibular third molars requiring osteotomy and odontosection [114]. Patients received capsules containing unspecified valerian extract (100 mg) or midazolam (15 mg) orally 1 h before surgical procedures. Heart rate, blood pressure, and respiratory rate were significantly lower when patients had taken midazolam compared to valerian, and no statistically significant differences in oxygen saturation were observed. These results showed that valerian root preparations have the potential to provide patients with the relaxation required without sedation and less somnolence than midazolam.

Recently, in a randomized, triple-blind clinical trial [115], 76 patients (17 men and 19 women each in the valerian and placebo groups, all candidates for coronary artery bypass graft surgery) were included. Patients in the valerian group began taking capsules containing 530 mg of valerian root extract, and the placebo group began taking capsules containing wheat flour the third night after surgery, 2 h before sleep for 30 nights after surgery. The results indicate that the applied valerian root extract could significantly improve the quality of sleep of patients after coronary artery bypass graft surgery.

The data from the above clinical tests suggest that their results can also be interpreted in the context of the GABAergic action of the constituents of the valerian root. Note that all published menopause-related studies, as discussed elsewhere [116] are inconsistent as they involve heterogeneous groups of women. Therefore, it is uncertain whether valerian preparations are really effective in alleviating central nervous system disorders in menopause.

Figure 1 represents typical compounds found in the described plants.

## 3. Conclusions

Many plants have the potential to alleviate various symptoms of neuropsychiatric origin. Therefore, they are used not only as registered remedies but also as dietary supplements, and there are many of these products on the market. Treatment with the plants presented in this review is particularly successful for milder forms of neurological disorders. Serious adverse effects, including memory loss or breakdown of psychometric functions, have not been associated with their use. However, according to the criteria for scientific knowledge of the pharmacological properties of these plants demonstrated in nonclinical and clinical trials, more studies are needed to validate the mechanism of action and identify the compounds responsible for these effects. With this knowledge, the plants described in this article will be used even more successfully and safely to alleviate or eliminate various mental disorders.

## Figures and Tables

**Figure 1 molecules-27-06021-f001:**
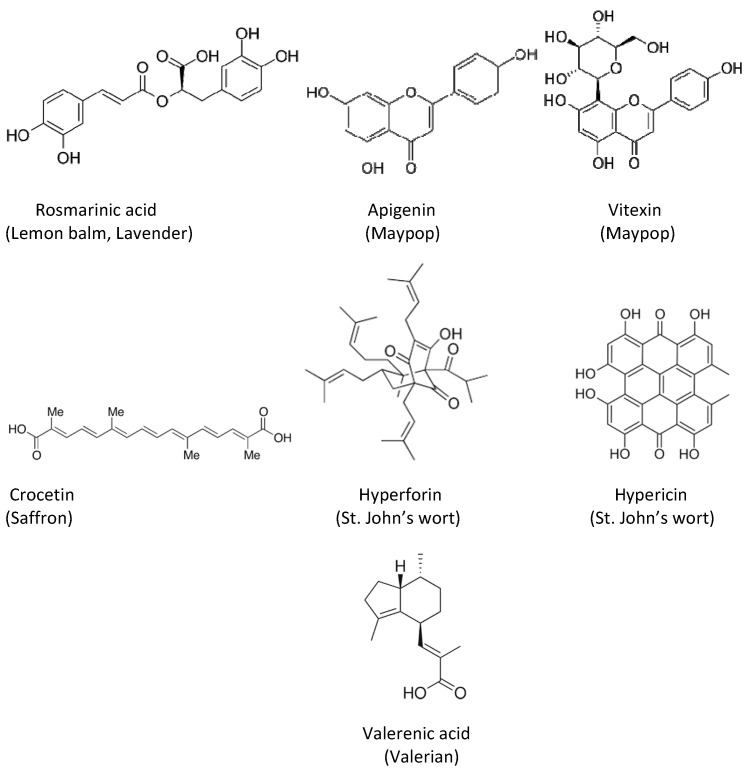
Typical constituents of plants used for the relief of mental disorders.

## Data Availability

Not applicable.

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
