# Peer review of "Medicinal Plants Used for Anxiety, Depression, or Stress Treatment: An Update"

_molecules, 2022, doi:10.3390/molecules27186021_

Round 1

Reviewer 1 Report

Dear Authors

The MS entitled "Medicinal plants acting on central nervous system: an update" was reviewed. the MS is very well written and designed. I am satisfied in its current form. I would like to propose an addition. Recently, Aconitum and Delphinium species have been well documented about its role in AD although, the species are toxic in nature however, the plants have been extensively studies against neuro degenerative diseases. Some of the literature could be found as.

https://www.sciencedirect.com/science/article/abs/pii/S0968089617301049 https://www.tandfonline.com/doi/abs/10.1080/10286020.2017.1319820 https://doi.org/10.3390/molecules27144348

Also, I suggest the authors provide some of the most prominent compounds in each plant species after discussion.

Author Response

"Dear Authors

The MS entitled "Medicinal plants acting on central nervous system: an update" was reviewed. the MS is very well written and designed. I am satisfied in its current form. I would like to propose an addition.

Recently, Aconitum and Delphinium species have been well documented about its role in AD although, the species are toxic in nature however, the plants have been extensively studies against neuro degenerative diseases. Some of the literature could be found as.

https://www.sciencedirect.com/science/article/abs/pii/S0968089617301049

https://www.tandfonline.com/doi/abs/10.1080/10286020.2017.1319820

https://doi.org/10.3390/molecules27144348

We thank the reviewer for the good opinion about the article. We would be happy to include any plant that has any effect on the central nervous system and is being intensively studied. However, the scope of such an article is far beyond the possibility of publication in one article. So, we decided to focus on “Plants and products derived from them that are commonly used in the Western world as dietary supplements or over-the-counter medicines for the above indications (the use of some of them is supported by the European Medicines Agency herbal monograph) were studied here, focusing on recent clinical trials, safety profile, and whether or not their use is justified.” as we wrote at the end of the introduction chapter.

Also, I suggest the authors provide some of the most prominent compounds in each plant species after discussion."

We highly appreciate the reviewer's comment. We have added Figure 1, where the most important  compounds in described plants are presented. Most plants contain a lot of different compounds, and only a few have been confirmed to have an effect on the nervous system. This is also the reason that these plants, despite their proven effectiveness, need to be further studied.

Reviewer 2 Report

The review entitled "medicinal plants acting on central nervous system: an update" is well written and structured starting with an introductive part followed by results (subdivided into several sections, one for each specie) and finally a short conclusion.

I would strongly recommend the following revisions before publication :

- the title is too general in view of the content of the review "acting on central nervous system" vs mainly depression, anxiety and stress.

- the methodology of the reviewing process should be explained (mainly the inclusion criteria of choosen articles but also the notion of "update")

- in the indroduction, authors should mentioned more deeply the recent review published in the field. There is a critical lack of references since several reviews mention the same plants or others.

Minor revision :

- p3, l134, check spelling of pipermethvstine

- double space must be checked line 179, 182, 349, 532, 626

- line 303, add a capital M to maypop

- line 566-568, please rephrase to avoid the repetition of severe/severe/severity

- line 574, is "decease" the correct word ?

- line 602, add a bracket after "Inventory"

Author Response

Reviewer 2:

The review entitled "medicinal plants acting on central nervous system: an update" is well written and structured starting with an introductive part followed by results (subdivided into several sections, one for eachspecie) and finally a short conclusion.

I would strongly recommend the following revisions before publication:

-the title is too general in view of the content of the review "acting on central nervous system" vs mainly depression, anxiety and stress.

We highly appreciate the reviewer's comment about the title of the manuscript. We changed it to “Medicinal plants used for anxiety, depression or stress treatment: an update”. This title will really be more appropriate given the focus we have followed in the article.

-the methodology of the reviewing process should be explained (mainly the inclusion criteria of choosen articles but also the notion of "update")

We highly appreciate the reviewer's comment. Due to the fragmentation of the field we decided to adopt a horizontal review strategy. We included plants that are already in dietary supplements or OTC drugs, which have been shown to be effective in alleviating mentioned mental disorders. We also included plants that have their own EMA (European Medicines Agency) monograph, which lists the effects on alleviating these problems. We also included those plants for which we have found recent data in the literature that have not been reviewed in previous reviews and systematic articles. A review of literature on plants with activity on mentioned mental disorders was conducted using PubMed, Google Scholar and plant regulatory authorities’ web pages (from 2015 till now). Publications in English, German and Slovenian were considered for screening. Since the journal does not require a chapter on Materials and Methods for review articles, we have not included it.

- in the indroduction, authors should mentioned more deeply the recent review published in the field. There is a critical lack of references since several reviews mention the same plants or others.

Thank you for this comment. Although references 2-4 already summarize the field of plants discussed in the article, we add a few more references to better cover previous reviews in this field (page 2, line 67).

Minor revision:

Many thanks to the reviewer for carefully reading the manuscript. We have corrected all errors found in the text, which follows in the rest of the text:

- p3, l134, check spelling of pipermethvstine

Pipermethvstine changed to pipermethystine

- double space must be checked line 179, 182, 349, 532, 626

The prefixes were arranged in mentioned lines accordingly.

- line 303, add a capital M to maypop

maypop changed to Maypop

- line 566-568, please rephrase to avoid the repetition of severe/severe/severity

The sentence was rephrased:

“Most subjects had severe (38.9%) or very severe (19.4%) RLS symptoms at enrollment, as inclusion criteria required symptoms occurring three times per week.”

- line 574, is "decease" the correct word?

Decease changed to decrease

- line 602, add a bracket after "Inventory" "

The bracket after "Inventory" was added.

Round 2

Reviewer 2 Report

The manuscript was rightly improved by the modifications, especially the title which is now more representative of the review subject. I thank the authors for having taken into account the remarks.